# Proteomic Analysis of Domestic Cat Blastocysts and Their Secretome Produced in an In Vitro Culture System without the Presence of the Zona Pellucida

**DOI:** 10.3390/ijms25084343

**Published:** 2024-04-14

**Authors:** Daniel Veraguas-Dávila, Camila Zapata-Rojas, Constanza Aguilera, Darling Saéz-Ruiz, Fernando Saravia, Fidel Ovidio Castro, Lleretny Rodriguez-Alvarez

**Affiliations:** 1Escuela de Medicina Veterinaria, Departamento de Ciencias Agrarias, Facultad de Ciencias Agrarias y Forestales, Universidad Católica del Maule, Km 6 Los Niches, Curicó 3340000, Chile; 2Department of Animal Science, Faculty of Veterinary Sciences, Universidad de Concepción, Av. Vicente Méndez 595, Chillan 3780000, Chile; mv.camilazapatarojas@gmail.com (C.Z.-R.); darlin.saez.r@gmail.com (D.S.-R.); fsaravia@udec.cl (F.S.); fidcastro@udec.cl (F.O.C.); llrodriguez@udec.cl (L.R.-A.); 3School of Veterinary Medicine, Faculty of Natural Sciences, San Sebastián University, Concepción 4081339, Chile; constanza.aguilera@uss.cl

**Keywords:** in vitro fertilization, felid embryos, zona pellucida, embryo–maternal communication, protein expression profile

## Abstract

Domestic cat blastocysts cultured without the zona pellucida exhibit reduced implantation capacity. However, the protein expression profile has not been evaluated in these embryos. The objective of this study was to evaluate the protein expression profile of domestic cat blastocysts cultured without the zona pellucida. Two experimental groups were generated: (1) domestic cat embryos generated by IVF and cultured in vitro (zona intact, (ZI)) and (2) domestic cat embryos cultured in vitro without the zona pellucida (zona-free (ZF group)). The cleavage, morula, and blastocyst rates were estimated at days 2, 5 and 7, respectively. Day 7 blastocysts and their culture media were subjected to liquid chromatography–tandem mass spectrometry (LC–MS/MS). The UniProt *Felis catus* database was used to identify the standard proteome. No significant differences were found in the cleavage, morula, or blastocyst rates between the ZI and ZF groups (*p* > 0.05). Proteomic analysis revealed 22 upregulated and 20 downregulated proteins in the ZF blastocysts. Furthermore, 14 proteins involved in embryo development and implantation were present exclusively in the culture medium of the ZI blastocysts. In conclusion, embryo culture without the zona pellucida did not affect in vitro development, but altered the protein expression profile and release of domestic cat blastocysts.

## 1. Introduction

Domestic cats are valuable models for the development of assisted reproductive techniques for the conservation of endangered felids [1,2,3]. In vitro embryo production techniques such as in vitro fertilization (IVF) and intracytoplasmic sperm injection (ICSI) have proven to be efficient tools for the conservation of wild felids, generating live offspring of different species [4,5,6]. However, the reduced population and the high incidence of teratospermia in wild felids limit the use of these techniques [7,8,9]. For this reason, interspecific somatic cell nuclear transfer (iSCNT) has been utilized to generate cloned embryos of endangered felids without using oocytes or sperm from these species [10,11,12]. The live birth of African wildcats (*Felis silvestris lybica*) and sand cats (*Felis margarita*) has been achieved by the transfer of cloned embryos generated by iSCNT using enucleated domestic cat oocytes as recipient cytoplasts [13,14,15]. However, the blastocyst rate and implantation capacity of felid embryos generated by iSCNT are reduced, and these embryos exhibit abnormal gene expression compared to embryos generated by SCNT and IVF [16,17,18]. To improve this low efficiency of SCNT, different methods involving the removal of the zona pellucida, such as handmade cloning and embryo aggregation, have been used for different species [19,20,21]. Previous studies have demonstrated that embryo aggregation significantly increases the blastocyst rate of cheetah (*Acinonyx jubatus)*, tiger (*Panthera tigris tigris*) and kodkod (*Leopardus guigna*) embryos generated by iSCNT [22,23,24]. However, the in vivo developmental capacity of these cloned embryos generated without the presence of the zona pellucida has not been evaluated.

In different species, the presence of an intact zona pellucida is not required during the in vivo development of in vitro-produced embryos after embryo transfer (ET) [25,26]. This fact has been proven by the live offspring generated after ET of cloned embryos produced by handmade cloning or embryo aggregation in bovines and equines [27,28,29]. However, it was postulated that an intact zona pellucida might be needed for the implantation of domestic cat embryos generated in vitro [5]. It was previously reported that domestic cat embryos transferred on day 7 at the hatching blastocyst stage were not able to implant, but morulae and blastocysts were [30]. More recently, it was reported that domestic cat blastocysts generated by IVF and in vitro cultured without the zona pellucida were not able to be implanted after ET [31,32]. Furthermore, altered expression of pluripotency (*SOX2* and *NANOG*) and differentiation genes (*YAP1* and *EOMES*) was observed in domestic cat blastocysts cultured without the zona pellucida [32,33]. Similar results have been described in mouse embryos: in vitro culture without the zona pellucida modified the gene expression pattern and reduced the implantation capacity of mouse embryos [34]. Recently, it was found that domestic cat blastocysts cultured without the zona pellucida had an altered expression of several miRNAs, which may indicate that their proteome could be affected [35]. However, more studies are needed to understand why domestic cat embryos cultured without the zona pellucida cannot be implanted.

Gene expression analysis does not predict protein abundance or provide information related to protein function [36]. Proteomic analyses of preimplantation embryos have been performed to identify new biomarkers of development and viability [37]. The proteome of a specific cell type is constantly altered by internal and external stimuli [38]. Moreover, the secretome is defined as proteins produced by all cell types and released at any time or under certain conditions [39]. Several types of biological fluids, including conditioned media, have been analyzed to identify specific proteins involved in disease states or cellular progression [40,41]. Numerous studies indicate that proteins released by embryos and the uterus are involved in early embryo−maternal dialogue and that this protein expression pattern constantly changes during embryo development [38,42]. Viable in vitro-produced embryos have a unique proteome, and some of these proteins are secreted into the culture medium, contributing to the secretome [38]. Furthermore, the pattern of proteins released by preimplantation embryos changes depending on in vitro culture conditions [43]. It has been reported that porcine embryos generated by IVF have an altered proteome, which induces aberrant embryo–maternal crosstalk [44]. These embryos had an increased abundance of energy metabolism- and proliferation-related proteins and a decreased abundance of methyl metabolism-related proteins [44]. Similarly, bovine embryos generated by IVF have an altered proteome compared to in vivo-produced embryos [45]. In vivo-produced embryos showed an enrichment in carbohydrate metabolism and cytoplasmic cellular components, while proteins found to be more abundant in IVF embryos were mostly localized in the mitochondrial matrix and involved in ATP-dependent activity [45]. In summary, acquiring information about embryo–maternal interactions will enable the design of new strategies to reduce early gestational losses. Proteomic analysis of in vitro-produced embryos and their secretome might provide important information regarding their reduced implantation capacity. For these reasons, the objective of this study was to evaluate the proteome and secretome of domestic cat blastocysts cultured without the zona pellucida.

## 2. Results

### 2.1. Experiment 1: In Vitro Development and Proteomic Analysis of Domestic Cat Blastocysts Cultured with (ZI) and without (ZF) the Zona Pellucida

#### 2.1.1. In Vitro Development of Domestic Cat Embryos from the ZF and ZI Groups

Domestic cat embryos were generated by IVF and in vitro cultured until the blastocyst stage. FBS was not added during IVC at any stage of embryo development. Embryos were cultured with the zona pellucida (zona intact (ZI)) or without the zona pellucida (zona-free group (ZF)).

The cleavage and morula rates were estimated at days 2 and 5 of IVC when these embryos were cultured in SOF-B. On day 5, morulae were cultured in M199-IVC, and the blastocyst rate was estimated on day 7. No significant differences were observed in the cleavage, morula, or blastocyst rates between the ZF and ZI groups (*p* > 0.05) (Table 1). These results indicate that domestic cat embryos can develop in vitro without FBS supplementation. Furthermore, under these conditions, in vitro culture without the zona pellucida did not affect embryo development until the blastocyst stage (Figure 1).

#### 2.1.2. Morphological Evaluation of Domestic Cat Blastocysts

The diameter (mean ± SD) of the ZF blastocysts (304.4 ± 121.9 µm) was significantly greater than that of their ZI counterparts (241.2 ± 70.3 µm) (*p* < 0.05). This might have been caused by a lack of growth limitation due to the absence of the zona pellucida in the ZF group. However, this blastocyst growth was not related to an increase in the total cell number and might have been caused by an increase in blastocoel expansion. No significant difference was found in the total cell number (mean ± SD) between blastocysts from the ZF (219.4 ± 98.9) and ZI groups (206.0 ± 54.3) (*p* > 0.05), which indicates that the in vitro culture method used in this study for ZF embryos had not affected the morphological quality of domestic cat blastocysts at day 7 of IVC (Figure 2).

#### 2.1.3. Proteomic Analysis of Domestic Cat ZF and ZI Blastocysts

##### Protein Identification

Proteomic analysis was performed using three replicates in each experimental group (ZF group: ZF01, ZF02, and ZF03; ZI group: ZI01, ZI02, and ZI03); each replicate contained 10 blastocysts from day 7 pooled together. Principal component analysis (PCA) is shown in Appendix A.

In this report, a total of 1604 proteins were identified in the ZF replicates, 443 of which were shared among the replicates. Moreover, in the ZI group, a total of 482 proteins were identified, and 142 proteins were shared among the samples. A total of 456 proteins were shared between blastocysts from the ZF and ZI groups (Figure 3).

##### Differential Expression Analysis

In this study, due to the limited concentration of some proteins present in blastocysts, only 250 of the identified proteins could be quantified. The IDs, logFC values and *p* values of all the quantifiable proteins in blastocysts are shown in the supplementary material (Appendix A). The expression analysis identified 42 differentially expressed proteins (DEPs) between blastocysts from the ZF and ZI groups (false-discovery rate, FDR < 0.05). Each comparison of the standardized protein abundance among the replicates and experimental groups is shown in the heatmap (Figure 4). Among the DEPs, 22 proteins were upregulated and 20 were downregulated in blastocysts from the ZF group compared to their ZI counterparts (*p* < 0.05), as shown in the volcano plot (Figure 5). The functions associated with embryo development of upregulated and downregulated proteins are shown in Table 2 and Table 3, respectively.

#### 2.1.4. Functional Classification of Differentially Expressed Proteins from the ZF and ZI Blastocysts

DEPs were functionally classified in GO: slim molecular function, biological process, and cellular component. In the molecular function classification, a high proportion of the upregulated proteins was involved in binding and catalytic activity. Similarly, in the downregulated group, a high proportion of proteins was involved in binding, but also many proteins were classified in structural molecule activity. This indicates that ZF blastocysts have a deregulated pattern of proteins associated with the binding of different molecules that could be important for embryo uptake and transcription: carbohydrate binding GO:0030246 (LGALS1), calmodulin binding GO:0005516 (EEF1A1), metal ion binding GO:0046872 (COX5B, UQCRFS1), heat-shock protein binding GO:0031072 (HSPA8, HSPA5), actin binding GO:0003779 (EZR), and RNA binding (HSP90B1, HNRNPH1, HDLBP, RNPS1, TRA2B, EMG1), among others. Furthermore, these blastocysts showed dysregulation of catalytic activity (ALDOA, ATP5PF) and a downregulation of proteins that contributed to structural integrity (RPL8, RPL18, RPL12, RPS25, RPL36A, COL1A2). Similarly, in the biological process classification, the ZF blastocysts showed an upregulated pattern of proteins involved in different cellular and metabolic processes important for embryo development: regulation of gene expression GO:0010468 (IGF2BP1), negative regulation of apoptotic process GO:0043066 (HSP90B1, HSPA5), glycolytic process GO:0006096 (ALDOA), and embryonic processes involved in female pregnancy (TLE6). However, these embryos also were affected by the downregulation of proteins directly involved in embryo development: embryo development GO:0009790 (TRA2B), blastocyst development GO:0001824 (EMG1), mitotic nuclear division GO:0140014 (PLK1), and cellular senescence (HMGA1). The pie charts show generic annotations of DEPs (Figure 6). A detailed functional classification (GO) is included in Appendix A. A detailed classification of the DEPs shown in Figure 6 is shown in Appendix A. A detailed GO functional classification of all the proteins identified in blastocysts is included in Appendix A.

### 2.2. Experiment 2: Proteomic Analysis of the Secretome of Domestic Cat Blastocysts Cultured with and without the Zona Pellucida

#### 2.2.1. Protein Identification in Conditioned Culture Media of ZF and ZI Blastocysts

In this study, we analyzed liquid samples of conditioned culture media collected from in vitro-produced blastocysts from each experimental group. The LC–MS/MS methodology used in this research was able to detect proteins in a minimal volume of culture medium. However, a minimal quantity of proteins was identified in these samples. In the ZF group, 30 different proteins were identified in the culture medium samples, but only 12 of these proteins were present in all the replicates. Similarly, in the ZI group, 39 proteins were identified in the culture medium samples, but only 13 were present in all the replicates. Therefore, 25 proteins were present in culture medium samples of blastocysts from the ZF and ZI groups. However, no significant differences were found in the expression of these proteins between the groups (*p* > 0.05). Additionally, 14 proteins were exclusively identified in the culture medium of the ZI blastocysts, and 5 proteins were detected only in the culture medium of the ZF blastocysts (Figure 7).

#### 2.2.2. Functional Classification of Proteins Identified in the Conditioned Culture Media

Despite the small number of identified proteins, our results indicate that 25 of these proteins were secreted or released by domestic cat blastocysts from the ZF and ZI groups. These proteins have different functions that might be important for embryo development and communication: keratins (KRT3, KRT10, KRT13, KRT15, KRT17, KRT74, KRT76, and KRT89), components of the extracellular matrix (hemicentin 1), binding and transport proteins (albumin, transferrin, calreticulin, fatty acid-binding protein 5, calmodulin 2, and prothymosin alpha), catalytic activity enzymes (cationic trypsin, cathepsin B, pantetheinase, and triosephosphate isomerase), cytoskeleton and microtubule organization proteins (CEP162, and β-actin), a plasma membrane receptor (GFRA3), a heat-shock protein (HSPA8) and one unnamed intermediate filament protein (ID: M3VUG4). Full information of protein shared by ZF and ZI samples is available in Appendix A.

Additionally, 14 proteins were identified only in samples from the ZI group, and we can assume that only ZI blastocysts were able to release these proteins, which are involved in different processes: structural and molecular stability: histone H2B (H2BC17), and H1-5; keratins: KRT32, KRT82 and KRT86; binding and transport proteins: annexin A2 (ANXA2); cell adhesion and junction organization: junction plakoglobin (JUP), plakophilin 1 (PKP1), desmoplakin (DSP), and FHL1; a multipass membrane protein: proteolipid protein 2 (PLP2); an enzyme: leucine aminopeptidase 3 (LAP3); and two nuclear membrane proteins: lamin A/C and SUN2. Furthermore, five proteins were identified exclusively in the culture medium of the ZF blastocysts, which indicates that these proteins are released only by domestic cat blastocysts without the zona pellucida. These proteins are involved in catalytic and oxidoreductase activity (FTO, LDHA, and GLUD1), programmed cell death (protein Mdm4), and binding and transport (metallothionein). GO of proteins identified in culture medium samples from the ZF and ZI groups are shown in Appendix A. Additionally, the functions associated with embryo development of proteins identified in culture media are summarized in Table 4. The functional classification is shown in Figure 8. GO annotation of all the proteins identified into culture medium samples is shown in Appendix A. Additional information for the pie charts (Figure 8) is shown in Appendix A.

## 3. Discussion

We previously reported that domestic cat blastocysts cultured without the zona pellucida cannot implant after ET, possibly because of the role of the zona pellucida in embryo–maternal crosstalk [32]. In this study, we observed an altered expression pattern of several proteins in domestic cat blastocysts cultured without the zona pellucida. Furthermore, compared with their ZI counterparts, domestic cat ZF blastocysts released different types of proteins into the culture media. Several of the identified proteins have been shown to be involved in different processes of embryo development in previous reports.

Among the proteins identified in the ZF blastocysts, three types of heat-shock proteins were upregulated (HSPA5, HSPA8, and HSP90B1). HSPA5 has been classified as a stress marker during embryo development, and a decrease in its expression has been associated with enhanced in vitro development of bovine embryos [66]. HSPA8 has been detected in oviductal fluid and histotrophs and has been associated with enhanced cell differentiation and fertilization [46,57,58]. The expression of IGF2BP1, a protein involved in developmental processes, was also upregulated in the ZF blastocysts. The expression of IGF2BP1 is significantly lower in murine blastocysts produced by IVF and SCNT than in in vivo-produced embryos [116]. Furthermore, overexpression of IGF2BP3 inhibited endometrial decidualization by downregulating TGF-β1, IGF2BP1 and IGF2BP2, which are homologous molecules of IGF2BP3 and might have similar functions [117,118]. Another upregulated protein was TLE6, which was overexpressed in ZF cat blastocysts. The TLE6 gene is expressed in oocytes and early embryos and is a member of the subcortical maternal complex (SCMC) [119,120]. The SCMC plays a physiological role in meiotic spindle positioning, mitochondrial redistribution, translation regulation, and zygotic epigenetic reprogramming [119,120]. Mutations in TLE6 are related to female infertility caused by developmental arrest and embryonic lethality [68,69,70].

The expression pattern of TLE6 in oocytes increases from the germinal vesicle to the metaphase II stage and subsequently decreases at the morula and blastocyst stages [121]. For this reason, an increase in the protein expression of TLE6 must be considered an alteration. However, more studies are needed to verify whether this increased expression affects the development of domestic cat blastocysts. Finally, ezrin (EZR) was another upregulated protein identified in the ZF blastocysts. EZR, together with radixin and moesin, forms the ERM complex, which is a regulator of microvillus formation in mammalian cells [59]. Several studies have reported that the EZR is related to the density of microvilli on the surface of preimplantation embryos and participates in oocyte maturation, fertilization, and preimplantation development [122,123,124]. Increased expression of EZR has been associated with an abnormally high density of microvilli in embryos produced by SCNT [125]. It is possible that abnormal upregulation of EZR affects the development of domestic cat ZF blastocysts.

Several types of proteins were downregulated in domestic cat ZF blastocysts, some of which are involved in trophoblast invasion and immunomodulation (HMGA1 and LGALS1), while others are involved in embryo development and progression (PLK1 and EMG1). HMGA1 is highly expressed in undifferentiated cells, such as embryonic cells [126]. HGA1 has an immunosuppressive role, and HGA1 overexpression has been associated with decreased expression in neutrophils and Th17 cells, but HGA1 expression increases when HMGA1 is methylated [91]. Furthermore, HGA1 has been associated with the invasion and proliferation of cancer cells by inducing genes involved in epithelial–mesenchymal transition (EMT) [127,128]. HMGA1 is expressed exclusively in trophoblast cells, and its immunosuppressive and proliferative effects are favorable for trophoblast invasion during placentation [127]. For this reason, the downregulation of HGA1 might be related to the reduced implantation capacity of domestic cat ZF blastocysts. LGALS1 or galectin 1 is involved in the modulation of the maternal immune response, fetomaternal tolerance, angiogenesis, implantation, and placentation [71,129]. Galectin 1 is also related to trophoblast migration, invasion, syncytium formation and expression of non-classical MHC class I molecules (HLA-G) [71]. It has been reported that human embryos secrete galectin 1 into the culture medium [71]. Furthermore, galectin 1 binds to mucin 1 on glandular epithelial cells and endometrial epithelial apical surface tissue, which indicates a possible role for it in blastocyst attachment [71,130]. Therefore, downregulation of galectin 1 might affect maternal recognition and implantation of domestic cat ZF blastocysts. PLK1 is a member of the four mammalian Polo-like kinases and participates in DNA replication and damage repair and during all stages of mitosis, including centrosome maturation, bipolar spindle formation, chromosome segregation, and cytokinesis [131,132]. Furthermore, PLK1 promotes the G2/M transition by activating the Cdk1–cyclin B complex [133]. PLK1 plays an important role in embryo development, and PLK1-/- knockout mouse embryos fail to survive after the eight-cell stage, indicating that PLK1 is important for cell cycle progression in preimplantation embryos [80]. Downregulation of PLK1 might affect the mitosis progression and development of ZF blastocysts. Essential mitotic growth 1 (EMG1) is a conserved nucleolar protein important for ribosome biogenesis [134]. In mouse embryos, EMG1 is critical for preimplantation development and is detected at the morula stage, and its expression increases in the inner cell mass at the blastocyst stage [79]. Emg1-/- mouse embryos suffer developmental arrest prior to the blastocyst stage and exhibit defects in ICM formation and nucleogenesis [79]. In the present study, the in vitro development of domestic cat ZF blastocysts was not affected. However, the downregulation of EMG1 might affect ICM formation, which could be related to the reduced expression of SOX2 and NANOG previously reported in these embryos [32].

Finally, proteins released by the ZF and ZI blastocysts were identified in the culture media. However, in both experimental groups, only a low quantity of proteins was detected, and it was not possible to perform a differential expression analysis. This could be because blastocysts were cultured without any protein supplementation 12 h prior to media collection, and their cells could have been in a quiescent stage, affecting their protein expression and secretion. Several studies have reported how serum or protein deprivation induces G0/G1 arrest in in vitro-cultured somatic cells of felids [135,136,137]. However, we observed that different types of specific proteins were released by the ZF and ZI blastocysts. Several keratins were identified in the culture media of the ZF and ZI blastocysts, and these proteins were specific to the *Felis catus* proteome. At present, several studies have reported that different types of keratins are important in cancer and embryo development [138]. Keratins have similar coiled-coil structural folds, but exhibit distinct surface chemistries, which enable diverse roles for keratins in extra- and intracellular functions that are essential during embryo development [109,139]. Furthermore, keratins reportedly play an immunoregulatory role via communication with immune cells [138]. Other important proteins include trypsin, which is important for hatching; transferrin, which has a chelating effect; and calmodulin 2, which is involved in embryonic division [140,141,142,143]. In this study, different enzymes, such as LDH and alpha-ketoglutarate-dependent dioxygenase, which are important for embryo metabolism and development, were detected in the medium of ZF blastocysts [92,144]. However, important proteins were present only in the culture medium of the ZI blastocysts, such as KRT32, KRT82 and KRT86 proteins. Keratins form intermediate filaments (IFs) of the epithelial cell cytoskeleton and are abundant in cytoplasmic projections of cumulus cells that cross the zona pellucida and contact the oocyte [109,145]. This explains the major proportions of keratins in the ZI samples. Furthermore, in mice, KRT86 is upregulated in the uterus during implantation, which indicates that this keratin might be involved in embryo implantation [99]. Other proteins important for cell adhesion and embryo implantation were identified in ZI blastocysts: desmoplakin, plakophilin 1, annexin A2 and FHL1 [100,115,146,147].

## 4. Materials and Methods

All chemical reagents were purchased from Sigma Aldrich Chemicals Company (St. Louis, MO, USA), except for those otherwise indicated.

### 4.1. Ethics Statement

All procedures involving animal manipulation were approved by the ethics committee (Comité de Bioética de la Facultad de Ciencias Veterinarias de la Universidad de Concepción, certificate of approval CBE-08-2020). Healthy female and male domestic cats aged between 6 months and 5 years were selected as oocyte and sperm donors for in vitro embryo production.

### 4.2. Experimental Design

Two experimental groups were used: (1) domestic cat embryos were generated by IVF and cultured in vitro (ZI group). (2) domestic cat embryos were generated by IVF and cultured in vitro without the zona pellucida (ZF group). Ovaries from domestic cats were collected by ovariohysterectomy. Immature cumulus–oocyte complexes (COCs) were collected from these ovaries by slicing and then subjected them to in vitro maturation (IVM). Subsequently, the in vitro-matured COCs were subjected to IVF using epididymal sperm. In the ZF group, the zona pellucida of presumptive zygotes was removed. Then, the ZF embryos were cultured in vitro using the well-of-the-well (WOW) system, while the ZI embryos were cultured normally in four-well dishes. The embryos were cultured for seven days. The cleavage, morula and blastocyst rates were evaluated. The total cell number and diameter of the blastocysts were estimated. Finally, day 7 blastocysts (experiment 1) or their culture media (experiment 2) were subjected to liquid chromatography–tandem mass spectrometry (LC–MS/MS). For this purpose, protein digestion was performed using trypsin. Then, 200 ng of the peptides was injected into a nanoUHPLC instrument (nanoElute, Bruker Daltonics, Billerica, MA, USA) coupled with a timsTOF-Pro mass spectrometer (Bruker Daltonics). The data were analyzed using MSFragger 3.5 and Plataform fragpipe v18.0 software, and the *Felis catus* database from UniProt was used to identify the standard proteome (Figure 9).

### 4.3. Ovariohysterectomy and Cumulus–Oocyte Complex (COC) Collection

Ovaries from female domestic cats were collected by ovariohysterectomy. The anesthesia protocol previously described by our research group was used in this study [148,149]. Immature COCs were recovered from ovaries by slicing in a 100 mm petri dish containing 10 mL of medium 199 with Earle’s salts supplemented with 0.18 mM HEPES, 5% FBS and 50 μg/mL gentamycin (He199) at 38.5 °C. Only grade I and II immature COCs were subjected to IVM.

### 4.4. In Vitro Maturation

In vitro maturation of immature COCs was performed in four-well dishes containing 500 µL of medium 199 with Earle’s salts supplemented with 0.3% fraction V BSA, 0.1 IU/mL FSH-LH (Pluset, Serono, Italy), 1 μg/mL 17β-estradiol, 0.36 mM sodium pyruvate, 2 mM glutamine, 2.2 mM calcium lactate, 20 ng/mL EGF, 10 µL/mL ITS (insulin, transferrin, selenium; Gibco Thermo Fisher Scientific, Waltham, MA, USA) and 50 μg/mL gentamycin (IVM-199) in a humidified gas atmosphere with 5.0% CO_2_ at 38.5 °C for 24–26 h.

### 4.5. Sperm Collection and In Vitro Fertilization

Only male domestic cats older than 10 months of age were used as sperm donors. The anesthesia procedure was the same as that described for ovariohysterectomy. Additionally, lidocaine (Lidocalm, 2%, Dragpharma, Santiago, Chile) was administered in the genital area as local anesthesia. Testes were transported in sterile 0.9% NaCl solution with 0.1% gentamycin at room temperature. The caudal portions of the epididymis were cut into small pieces of ~1 mm in a 100 mm petri dish containing 10 mL of He199 supplemented with 0.3% fraction V BSA instead of FBS (He199-BSA) at 38.5 °C. Epididymal spermatozoa were processed and stored at 4 °C, as we previously described [150].

IVF was conducted in four-well dishes with 500 µL of TALP medium supplemented with 6 mg/mL BSA, 0.36 mM sodium pyruvate, 1 mM glutamine, 2.2 mM calcium lactate, 1% MEM nonessential amino acids (NEAA), 0.5% MEM essential amino acids (EAA), 0.01 mg/mL heparin sodium salt and 50 μg/mL gentamycin (TALP-IVF). Refrigerated spermatozoa were allowed to swim up for 30 min in He199-BSA at 38.5 °C [150]. The pellet was collected and resuspended in TALP-IVF. For IVF, 20 to 30 COCs were co-incubated with 1.5 to 2.5 × 10^6^ spermatozoa/mL in a humidified atmosphere of 5% CO_2_ in air at 38.5 °C for 24 h. Subsequently, cumulus cells were removed from the presumptive zygotes using a 0.5 mg/mL hyaluronidase solution and vortexed for 6 min.

### 4.6. In Vitro Embryo Culture

After IVF, in the ZF group, the zona pellucida from presumptive zygotes was removed by incubation in 2 mg/mL of pronase for 4 min, at 37 °C. Subsequently, presumptive zygotes were washed three times in He199 supplemented with 30% FBS (He199-30) to remove pronase, and then washed three times in SOF medium to eliminate FBS.

In the ZI and ZF groups, presumptive zygotes were cultured in supplement SOF medium [151]. However, FBS was replaced by ITS (10 µL/mL) to avoid the presence of contaminating bovine proteins. The culture was performed in four-well dishes containing 500 µL of SOF medium supplemented with 0.37 mM trisodium citrate, 2.77 mM myo-inositol, essential and nonessential amino acids (final concentration 1×), 50 µg/mL gentamycin, 10 µL/mL ITS, 20 ng/mL EGF, and 3 mg/mL essentially fatty acid-free BSA (SOF-B). In the ZI group, 10 to 20 embryos were placed into each well. On the other hand, ZF embryos were cultured in four-well dishes using the WOW system [152]. In the WOW system, several microwells were created in the culture dish, and each ZF embryo was individually cultured in one microwell to prevent the disaggregation of blastomeres [152]. In this study, the microwells were created using an aggregation needle (BLS Ltd, Budapest, Hungary. aggregation needles DN-09N). Eighty microwells were created in a four-well dish, and up to 20 ZF embryos were cultured per well. In addition, 500 µL of SOF-B was added per well, and mineral oil was not used to avoid oil traces in the mass spectrometry analysis. In this study, handmade microwells were used instead of commercial microwells, because we previously obtained consistent in vitro development of domestic cat embryos generated by IVF and SCNT using these handmade microwells for IVC [22,32].

In both experimental groups, at day 2 of IVC cleavage, embryos were selected and the rest were discarded. At day 5 of IVC, morulae were selected, the remaining embryos were discarded, and SOF-B medium was replaced by supplemented medium 199. Morulae were cultured in medium 199 with Earle’s salts supplemented with 0.37 mM trisodium citrate, 2.77 mM myo-inositol, essential and nonessential amino acids (final concentration 1×), 50 µg/mL gentamycin, 3 mg/mL essentially fatty acid-free BSA, 10 µL/mL ITS and 20 ng/mL EGF (M199-IVC). The culture was conducted in a humidified atmosphere of 5% CO_2_, 5% O_2_ and 90% N_2_ at 38.5 °C for seven days. The cleavage, morula and blastocyst rates were estimated at days 2, 5 and 7, respectively.

### 4.7. Sample Collection

In experiment 1, day 7 blastocysts were collected from the ZI and ZF groups, and only good-quality blastocysts (grade A) with a distinguishable inner cell mass (ICM) and a trophoblast formed by numerous cells were selected. Three samples (biological replicates) were created for each experimental group. These samples consisted in 10 blastocysts (from their respective group) pooled together in a microcentrifuge tube. These samples were washed three times in cold PBS to eliminate the traces of BSA.

In experiment 2, to evaluate the secretome of ZI and ZF blastocysts, ten day 7 blastocysts were cultured together in one well of a 96-well culture plate with 50 µL M199-IVC without BSA for 12 h. The culture medium was collected and filtrated used a 3 kDa Amicon filters (Amicon ultra 0.5 mL—3 kDa) and eluted in 40 µL of cold PBS following the manufacturer’s instructions. This protein filtrate was considered one sample (biological replicate), and three biological replicates were created for both experimental groups.

### 4.8. Morphological Evaluation of Blastocysts

#### 4.8.1. Diameter Measurement

After in vitro culture, day 8 blastocysts from the ZI and ZF groups were imaged using a camera mounted on a stereomicroscope (Micrometrics CMOS Digital Camera), and the diameter of the blastocysts was measured from the captured images using Micrometrics SE Premium software 4.5.1.

#### 4.8.2. Total Cell Count

Blastocysts were fixed in a 3% glutaraldehyde solution for 72 h at 4 °C. Fixed blastocysts were stained with 5 μg/mL Hoechst 33342 for 20 min. The stained embryos were placed on a slide and covered with a coverslip. Visualization was achieved using the EVOS FL Auto Cell Imaging System (Thermo Fisher Scientific, Waltham, MA, USA) with a 20× magnification objective.

### 4.9. Proteomic Analysis

#### 4.9.1. Protein Extraction

Sample processing, LC–MS/MS, and data analysis were conducted by the MELISA Institute group. The samples were lyophilized and then resuspended in a pH 8 solution with 8 M urea and 25 mM NH_4_HCO_3_. Samples were homogenized by ultrasound with 10 s (on/off) pulses at 50% amplitude for 1 min. Subsequently, the processed samples were incubated on ice for 5 min, and the cellular debris was removed by centrifugation at 21,000× *g* for 10 min at 4 °C. Protein quantification was performed using the Qubit protein assay (# Q33212, Invitrogen, Thermo Fisher Scientific, Waltham, MA, USA).

#### 4.9.2. Protein Digestion for Mass Spectrometry

The obtained proteins were subjected to chloroform–methanol extraction. The samples were equilibrated at room temperature for 10 min and then centrifuged at 16,000× *g* for 15 min at 4 °C. The pellet was washed three times in cold 80% acetone and allowed to dry in a rotary concentrator.

The samples were resuspended in 30 µL of 8 M urea and 25 mM NH_4_HCO_3_. Subsequently, the samples were reduced with 20 mM DTT and incubated for 1 h at room temperature. Then, 20 mM iodoacetamide was added, and the samples were incubated for 1 h at room temperature in the dark. After that, the samples were diluted in a 25 mM NH_4_HCO_3_ solution.

Protein digestion was performed using MS-grade trypsin (#V5071, Promega, Madison, WI, USA) at a 1:50 (protease/protein; m/m) ratio for 16 h at 37 °C. The reaction was stopped by the addition of 10% formic acid. Then, the samples were subjected to Clean Up Sep-Pak C18 Spin Columns following the manufacturer’s instructions. Subsequently, the clean peptides were dried using a rotary concentrator at 1000 rpm and 10 °C overnight.

#### 4.9.3. Liquid Chromatography–Tandem Mass Spectrometry (LC–MS/MS)

Two hundred nanograms of peptides were injected into a nanoUHPLC (nanoElute, Bruker Daltonics, Billerica, MA, USA) coupled with a trapped ion mobility spectrometer—Quadrupole Time-of-Flight Mass Spectrometer (timsTOF Pro, Bruker Daltonics) using an Aurora UHPLC column (25 cm × 75 μm ID, 1.6 μm C18, IonOpticks, Fitzroy, Australia). Liquid chromatography was performed using a gradient of 2% at 35% tampon B (0.1% formic acid—acetonitrile) for 90 min. The results were collected using TimsControl 2.0 software (Bruker Daltonics), and 10 cycles of PASEF were used, with a mass range of 100–1700 *m*/*z*, 1500 V capillary ionization, 180 °C temperature, and a TOF frequency of 10 kHz, with a 50,000 FWHM resolution.

#### 4.9.4. Protein Identification

The data were analyzed by the software MSFragger 3.5 [153], using the platform Fragpipe v18.0 (https://fragpipe.nesvilab.org/; accessed on 13 September 2022), and the “default” workflow was used in a data analysis server composed of 48 nuclei and 512 Gb of RAM. Precursor mass tolerance parameters of −20 to 20 PPM and a mass fragment tolerance of 20 PPM Da were used. The database used for protein identification was the *Felis catus* standard proteome available in UniProt (40.213 total entries). FDR (false-discovery rate) estimation was included using a decoy database. We used a ≥1% FDR estimation and 1 minimal single peptide for protein identification.

#### 4.9.5. Protein Functional Classification

Differentially expressed proteins (DEPs) of blastocysts and the proteins detected in their conditioned culture media were functionally classified using the PANTHER18.0 classification system (https://www.pantherdb.org/; accessed on 15 February 2024) with its functional classification analysis option (PANTHER GO: slim molecular function, biological process, and cellular component). Not all the protein IDs were identified by the PANTHER18.0 software. For this reason, the GO annotation (slimming set) information of the UniProt website (https://www.uniprot.org/; accessed on 15 February 2024), and the ancestor chart of QuickGO (https://www.ebi.ac.uk/QuickGO) were used for functional classification and their data were added to the pie charts.

### 4.10. Statistical Analysis

#### 4.10.1. In Vitro Embryo Development and Morphological Analysis

The Wilcoxon nonparametric test was used to evaluate in vitro embryo development between the ZI and ZF groups. A *t* test was used to evaluate the diameter and total cell number in blastocysts between groups. The statistical software InfoStat (2020 version; University of Cordoba, Cordoba, Argentina) was used to evaluate significant differences (*p* < 0.05).

#### 4.10.2. Protein Quantification LFQ (Label-Free Quantification)

Using the results obtained from the protein identification process, we selected the columns corresponding to the access codes of UniProt. The resulting matrix was subjected to median normalization. The differentially expressed proteins were determined using a *t* test with the Benjamini–Hochberg multiple-correction test (*p* < 0.05). The identification of differentially expressed proteins (DEPs) was performed via ZF:ZI comparison.

## 5. Conclusions

In conclusion, culture of domestic cat blastocysts without the zona pellucida did not affect in vitro development. However, we detected altered expression and release of several proteins in the ZF blastocysts, which indicates that the zona pellucida plays a role in protein expression and release. Several of these proteins are involved in embryo development and implantation according to previous reports. However, additional studies are needed to determine whether the presence of the zona pellucida is critical for maternal embryo communication and implantation in domestic cats.

## Figures and Tables

**Figure 1 ijms-25-04343-f001:**
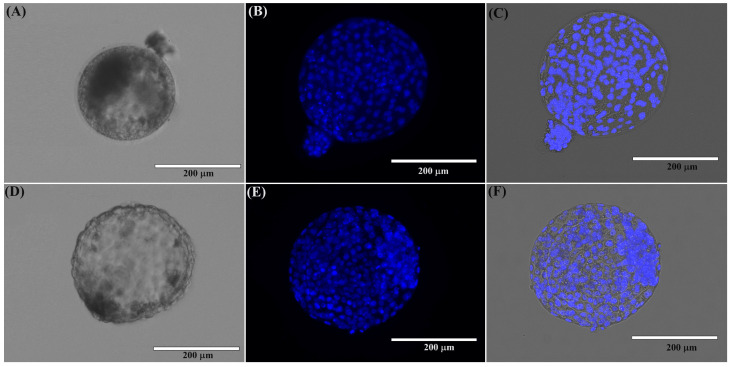
Domestic cat blastocysts. (**A**) Domestic cat blastocyst from the ZI group at day 7 of IVC (20X). (**B**) Domestic cat ZI blastocyst stained with Hoechst (20X). (**C**) ZI blastocyst merge (Hoechst and transmitted light) (20X). (**D**) Domestic cat blastocyst from the ZF group at day 7 of IVC (20X). (**E**) Domestic cat ZF blastocyst stained with Hoechst (20X). (**F**) ZF blastocyst merge (Hoechst and transmitted light) (20X).

**Figure 2 ijms-25-04343-f002:**
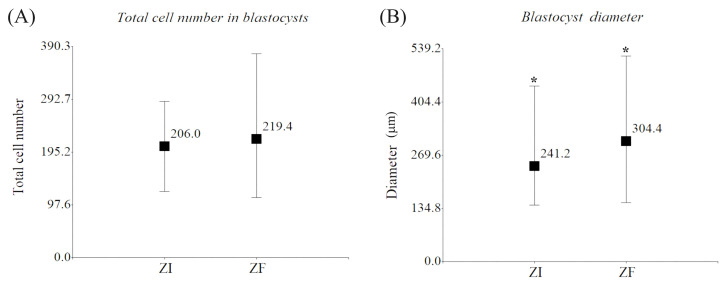
Morphological evaluation of blastocysts. (**A**) Total cell number in blastocysts (mean; min/max) from the ZI (206.0; 122/289) and ZF groups (219.4; 111/377). (**B**) Diameter (mean; min/max) of blastocysts from the ZI (241.2; 142.9/444.8 µm) and ZF groups (304.4; 149.8/520.3 µm). ***** Significant differences between bars, *p* < 0.05.

**Figure 3 ijms-25-04343-f003:**
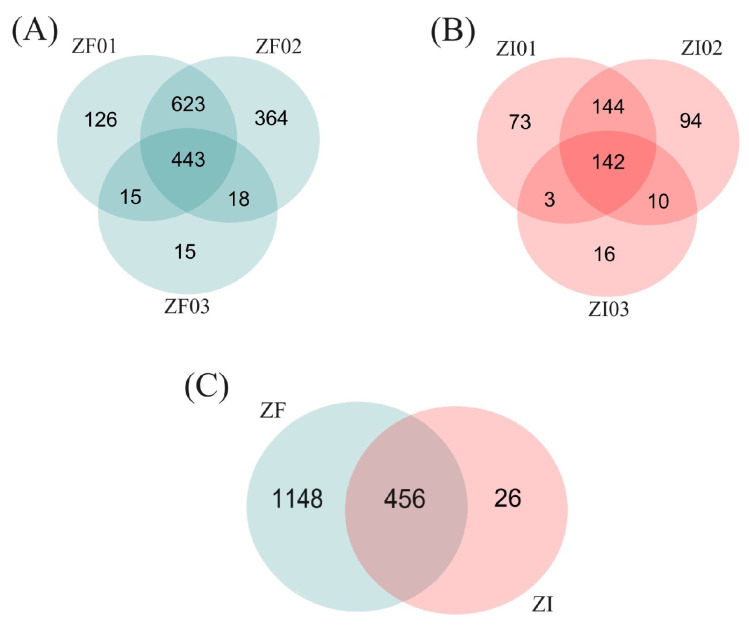
Venn diagrams of the proteins identified in the ZF and ZI blastocysts. (**A**) Comparison among the protein groups identified in blastocyst samples: ZF01, ZF02 and ZF03 (ZF group). (**B**) Comparison among the protein groups identified in blastocyst samples: ZI01, ZI02 and ZI03 (ZI group). (**C**) Comparison between the total proteins identified in the ZF and ZI groups. In sum, 456 proteins were shared between groups: 1148 were exclusively detected in ZF blastocysts and 26 were expressed only by ZI blastocysts.

**Figure 4 ijms-25-04343-f004:**
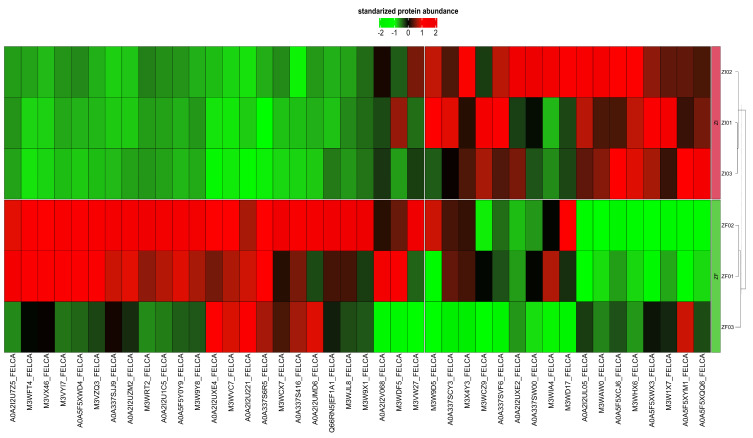
Heatmap of DEPs identified in the blastocysts (ZF vs. ZI). The 42 DEPs are grouped into blocks. Each block corresponds to proteins with similar expression patterns.

**Figure 5 ijms-25-04343-f005:**
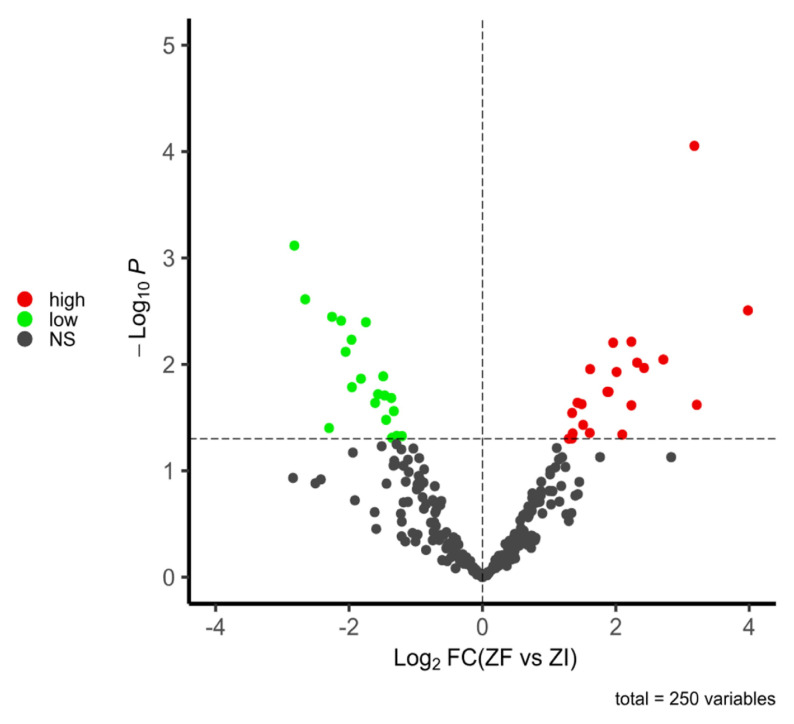
Volcano plot of the DEPs (ZF vs. ZI). In sum, 250 proteins were quantified: 22 were upregulated and 20 were downregulated (FDR < 0.05).

**Figure 6 ijms-25-04343-f006:**
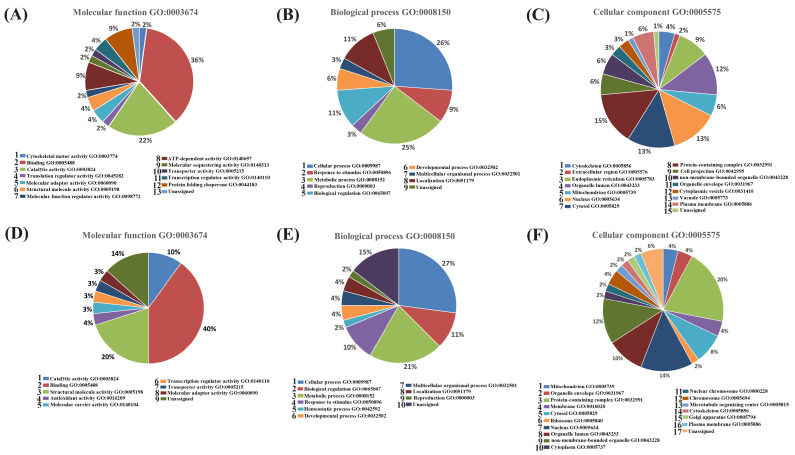
Functional classification (descriptive analysis) of DEPs identified between ZF vs. ZI blastocysts. Pie charts were created based on the functional classification of the PANTHER18.0 classification system, UniProt GO annotation, and QuickGO ancestor chart. Upregulated proteins (**A**–**C**). (**A**) Molecular function (GO:0003674), (**B**) biological process (GO:0008150), and (**C**) cellular component (GO:0005575). Downregulated proteins (**D**–**F**). (**D**) Molecular function (GO:0003674), (**E**) biological process (GO:0008150), and (**F**) cellular component (GO:0005575).

**Figure 7 ijms-25-04343-f007:**
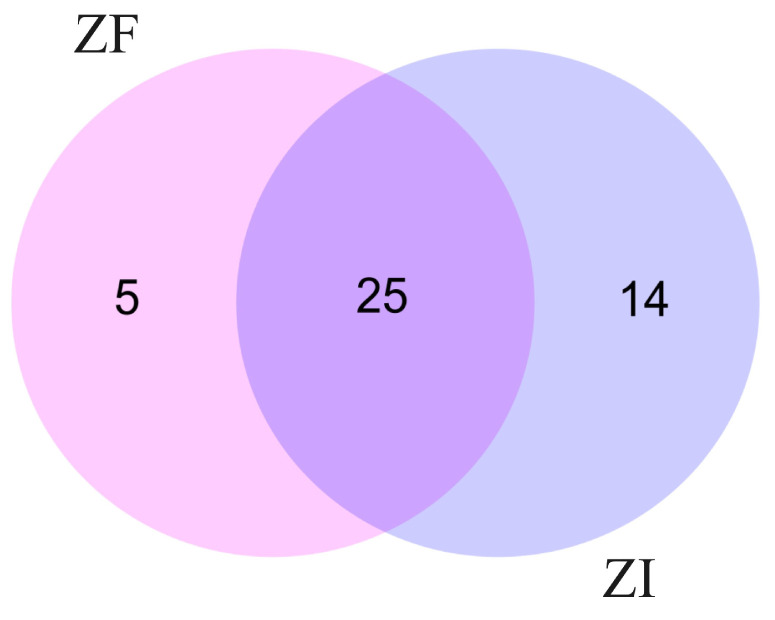
Venn diagram of proteins identified in the conditioned culture medium of blastocysts: 25 proteins were shared between ZF and ZI samples, 5 proteins were detected only in the culture medium of ZF blastocysts, and 14 in the culture medium of ZI blastocysts.

**Figure 8 ijms-25-04343-f008:**
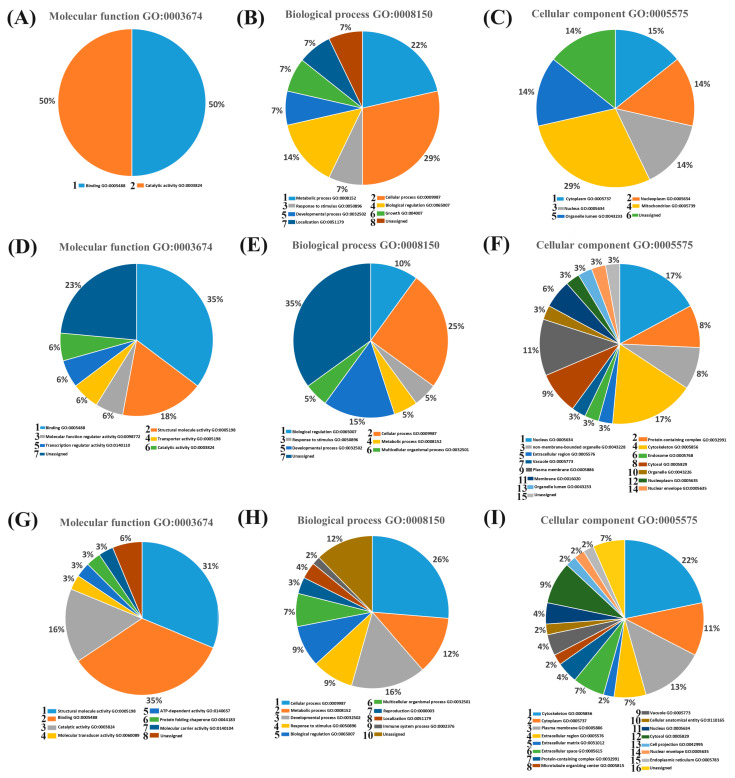
Functional classification (descriptive analysis) of proteins detected in the conditioned culture media of ZF and ZI blastocysts. Pie charts were created based on the functional classification of the PANTHER18.0 classification system, UniProt GO annotation, and QuickGO ancestor chart. (**A**–**C**) Proteins identified in the conditioned culture medium of ZF blastocysts. (**D**–**F**) Proteins identified in the conditioned culture medium of ZI blastocysts. (**G**–**I**) Proteins shared by the samples of conditioned culture media of ZF and ZI blastocysts.

**Figure 9 ijms-25-04343-f009:**
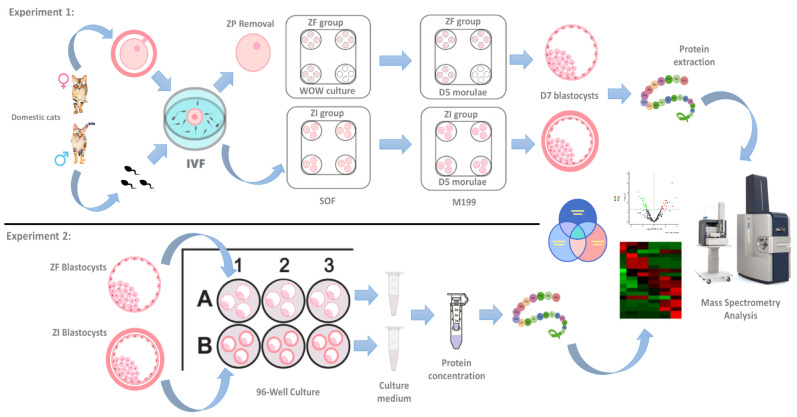
Experimental design. Workflow diagram of in vitro fertilization, embryo culture and proteomic analysis of day 7 blastocysts (experiment 1) and their conditioned culture media (experiment 2).

**Table 1 ijms-25-04343-t001:** In vitro development of domestic cat embryos cultured with (ZI) and without (ZF) the zona pellucida for seven days.

Groups	*n**	Total Oocytes	Cleavages (%Mean ± SD)	Morulae (%Mean ± SD)	Blastocysts (%Mean ± SD)	Hatching Blastocysts (%Mean ± SD)
ZI	10	351	140 (39.9 ± 11.6)	71 (50.7 ± 14.3)	44 (31.4 ± 8.5)	8 (5.7 ± 10.6)
ZF	10	470	201 (42.8 ± 15.8)	95 (47.3 ± 27.9)	62 (30.8 ± 17.8)	

*n**: number of replicates.

**Table 2 ijms-25-04343-t002:** Upregulated proteins detected in domestic cat blastocysts cultured without the zona pellucida, log_2_FC (ZF vs. ZI), and their function in embryo development.

Protein ID	Log_2_FC	Name	Function	References
A0A2I2U1C5_FELCA	3.98060042	MYH9	Fertilization, early development	[46,47]
M3W9X1_FELCA	3.21413703	PDIA4	Morula to blastocyst transition	[45]
M3VZQ3_FELCA	3.17734479	TUFM	Mitochondrial translation	[48]
M3VYI7_FELCA	2.71239201	IGF2BP1	m6A modification	[49]
A0A337S6R5_FELCA	2.42297848	ATP5PF	ATP production	[50]
A0A2I2UXE4_FELCA	2.31927503	HNRNPH1	Post-transcriptional regulation	[51]
A0A5F5XWD4_FELCA	2.23313382	HSP90B1	Stress and damage protection	[52]
A0A2I2U221_FELCA	2.23313259	CCT8	Oocyte sperm interaction, cell growth	[53,54]
M3WRT2_FELCA	2.0962291	RPL8	Ribosomal biogenesis	[55]
M3W9Y8_FELCA	2.01100493	RPL18	Ribosomal biogenesis	[56]
A0A2I2UZM2_FELCA	1.95964111	HSPA8	Cell differentiation, fertilization	[46,57,58]
A0A5F5Y0Y9_FELCA	1.89221244	EZR	Microvilli formation	[59]
M3WCX7_FELCA	1.87232765	EEF1B2	Ribosomal translation factor	[60]
M3VX46_FELCA	1.61502009	COX5B	Mitochondrial function	[61]
A0A2I2U7Z5_FELCA	1.60935135	BCAP31	Apoptosis, embryo implantation	[62]
EF1A1_FELCA	1.51020066	EEF1A1	Ribosomal translation factor	[60]
A0A2I2UMD6_FELCA	1.48842654	HDLBP	Anti-inflammatory, antioxidant	[63]
A0A337SJJ9_FELCA	1.42487437	RNPS1	Pre-mRNA splicing	[64]
M3WVC7_FELCA	1.35282905	YWHAZ	Cellular communication system	[65]
M3WFT4_FELCA	1.34473579	HSPA5	Stress marker	[66]
A0A337S416_FELCA	1.3425531	ALDOA	Glycolysis in oocytes and embryos	[67]
M3WJL8_FELCA	1.29586264	TLE6	Infertility, developmental arrest	[68,69,70]

**Table 3 ijms-25-04343-t003:** Downregulated proteins detected in domestic cat blastocysts cultured without the zona pellucida, log_2_FC (ZF vs. ZI), and their function in embryo development.

Protein ID	Log_2_FC	Name	Functions	References
M3W9D5_FELCA	−1.20664542	LGALS1	Blastocyst attachment	[71]
M3WIA4_FELCA	−1.28571479	COA3	Mitochondrial respiratory chain	[72]
M3X4Y3_FELCA	−1.32858383	UQCRFS1	Mitochondrial structural	[73]
A0A2I2V068_FELCA	−1.35619626	PSME3IP1	Transmembrane protein	[74]
M3WD17_FELCA	−1.36580332	RPL12	Protein synthesis, hatching	[75]
M3WDF5_FELCA	−1.44449596	TRA2B	mRNA splicing, endometrium	[76]
M3VW27_FELCA	−1.46532	RPS25	Repeated implantation failure	[77]
M3WAW0_FELCA	−1.48787899	RRP9	Morula to blastocyst transition	[78]
A0A2I2UL05_FELCA	−1.5655253	PRXL2A	Antioxidant, embryo–maternal crosstalk	[44]
M3W1X7_FELCA	−1.60814943	EMG1	Preimplantation development	[79]
M3WHX6_FELCA	−1.74681032	PLK1	Mitosis, embryo development	[80]
A0A5F5XWX3_FELCA	−1.82092779	KPNA2	Preimplantation embryo arrest	[81]
A0A5F5XQQ6_FELCA	−1.9562502	KHDC3L	Epiblast stability and viability	[82]
A0A337SCY3_FELCA	−1.9626275	RPL36A	Protein synthesis in oocytes and embryos	[83,84]
A0A337SW00_FELCA	−2.0509994	H2BC3	DNA repair in embryos	[85]
A0A5F5XCJ6_FELCA	−2.11846902	YIPF3	Golgi transport, ER function	[86]
M3WCZ9_FELCA	−2.25459988	H2BC18	Immunity	[87]
A0A5F5XYM1_FELCA	−2.29930597	COL1A2	Embryo extracellular matrix	[88]
A0A2I2UXE2_FELCA	−2.65692623	GNG12	Inflammation, immunity	[89,90]
A0A337SVF6_FELCA	−2.81974135	HMGA1	Immunosuppression	[91]

**Table 4 ijms-25-04343-t004:** Proteins exclusively identified in the conditioned culture media of ZF and ZI blastocysts.

Protein ID	Group	Name	Function	References
A0A337SNF5_FELCA	ZF	FTO	Epigenetic modulations	[92]
A0A337S839_FELCA	ZF	MDM4	p53 regulation, cell proliferation	[93]
A0A337S8L1_FELCA	ZF	LDHA	Embryo metabolism	[94]
A0A337S103_FELCA	ZF	MT1X	Metal ion homeostasis, stress response	[95]
M3W8Q7_FELCA	ZF	GLUD1	Glutaminolysis metabolism	[96,97]
A0A2I2USF3_FELCA	ZI	H2BC17	Regulation of embryo development	[98]
A0A5F5XCD9_FELCA	ZI	KRT86	Embryo implantation	[99]
A0A337SCD2_FELCA	ZI	ANXA2	Cell adhesion, embryo implantation	[100]
A0A2I2UBN8_FELCA	ZI	JUP	Cell adhesion, embryo morphology	[101]
M3W114_FELCA	ZI	PLP2	Gastrulation	[102]
A0A2I2UGZ3_FELCA	ZI	KRT32	Cell adhesion, differentiation, and migration	[103]
A0A5F5XSP5_FELCA	ZI	LAP3	Pregnancy biomarker	[104,105]
M3W2I5_FELCA	ZI	PKP1	Desmosome stabilization and maturation	[106]
M3WCK3_FELCA	ZI	DSP	Desmosome formation, trophoblast stabilization	[107,108]
M3VUF5_FELCA	ZI	KRT82	Intermediate filament formation	[109]
M3W584_FELCA	ZI	H1-5	Differentiation of pluripotent cells	[110,111]
A0A5K1VP01_FELCA	ZI	LMNA	Cell plasticity and differentiation	[112,113]
A0A337SS59_FELCA	ZI	SUN2	Mitosis, mitotic spindle formation	[114]
A0A2I2U7V8_FELCA	ZI	FHL1	Blastocyst–epithelial adhesion	[115]

## Data Availability

The data presented in this study are available upon request from the corresponding author.

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
