# Peer review of "Proteomic Analysis of Domestic Cat Blastocysts and Their Secretome Produced in an In Vitro Culture System without the Presence of the Zona Pellucida"

_ijms, 2024, doi:10.3390/ijms25084343_

Round 1

Reviewer 1 Report

Comments and Suggestions for Authors

General comment

The study has great scientific value. The research is innovative, using advanced research methods and the manuscript is well written and is suitable for publication International Journal of Molecular Science.

Minor revision

Paragraph 4.3 There is no need to describe the anesthetic procedure, I suggest deleting this description.

Paragraph 4.3 Just a comment - I was surprised that You don't use commercial microwell dishes, but handmade ones

Did You covered the microwells with medium only or also with oil?

Did You change the medium during culture?, I understood that you did not change the medium

Paragraph 4.7-  how did You select blastocysts to sample pooling? Did you pool random blastocysts or just top quality ones?

Paragraph 4.8 - You described successively  4.8.1. Total cell counting and then 4.8.2. Diameter measurement. Shouldn't it be the other way around? First measure the diameter and then fix and cell counting?

Author Response

Dear reviewer

I sincerely thank you for all your comments, corrections, and contributions to this study. I will proceed to respond to each one of your comments:

  1. Paragraph 4.3. Thank for this comment. The description of the anesthetic procedure was deleted, and the respective references were added instead. Lines 434-436

  1. Paragraph 4.3. Yes, we commonly have been used handmade microwells in our studies, the commercial microwells are not available in our country. Furthermore, we previously obtained consistent in vitro development of domestic cat embryos generated by IVF and SCNT using these handmade microwells for IVC [22,32]. This was added in Lines 486-490.

The microwells were not covered with mineral oil to avoid the possible traces of oil at the moment of mass spectrometry analysis. This also was added in the lines 486-487.

Yes, the supplemented SOF medium was changed at day-5 of IVC for supplemented m-199, only morulae were cultured in medium-199 until day-7. This was made to discard the possible proteins secreted by the arrested or less competent embryos during day 1-4 of IVC. Furthermore, we previously observed that changing SOF medium to medium-199 on day 5 of IVC improved the blastocyst rate when FBS was not supplemented (unpublished results). This was described in the previous version of the manuscript. Additionally, new information was added to this corrected version, Line 493.

  1. Paragraph 4.7. Yes, blastocysts selection was performed prior to sample pooling. Day-7 blastocysts were collected from the ZI and ZF groups, only good quality blastocysts (grade A) with a distinguishable inner cell mass (ICM) and a trophoblast formed by numerous cells were selected. This information was added to Lines 501-503.

  1. Paragraph 4.8. Indeed, diameter measurement of blastocysts was performed before total cell counting. The redaction of the paragraph 4.8 was corrected in Lines 516-526.

Reviewer 2 Report

Comments and Suggestions for Authors

In present study Veraguas-Dávila with co-authors demonstrate that zona pellucida plays a prominent role in expression and release of some proteins which could be involved in embryo development and implantation, although did not affect in vitro development. The article can be published after minor revision; there are several remarks to the text, which are listed below:

1. It is necessary to change the names of the Results subsections, i.e., # 2.1. on line 99 and # 2.2. on line 207, since the headings "Experiment 1" and "Experiment 2" are not informative. In the first case, perhaps this would be something like "comprehensive analysis of zona-free and zona-intact blastocysts" or something like that.

2. In Fig. 6 (line 199) and Fig. 8 (line 257), there are some small and completely unreadable signatures. If it is not possible to change the format of the figure in such a way as to increase the font size, it is necessary to make numerical designations and disclose them in the caption to the figure.

3. In Figure 9 (line 408), it is necessary to add a second cat to the scheme since, in the current version, both sperm and oocytes are taken from the same animal.

Author Response

Dear reviewer

I sincerely thank you for all your comments, corrections, and contributions to this study. I will proceed to respond to each one of your comments:

  1. The text on line 99 was replaced by: Experiment 1: in vitro development and proteomic analysis of domestic cat blastocysts cultured with (ZI) and without (ZF) the zona pellucida. The text on line 207 was replaced by: Experiment 2: proteomic analysis of the secretome of domestic cat blastocysts cultured with and without the zona pellucida. In line 222.
  2. The text format of Fig.6 and Fig.8 was changed; the font size was increased.
  3. A second image of a cat was added to Fig.9.
